# Mapping Understory Vegetation Density in Mediterranean Forests: Insights from Airborne and Terrestrial Laser Scanning Integration

**DOI:** 10.3390/s23010511

**Published:** 2023-01-03

**Authors:** Carlotta Ferrara, Nicola Puletti, Matteo Guasti, Roberto Scotti

**Affiliations:** 1CREA, Research Centre for Forestry and Wood, Via Valle della Quistione, IT-00166 Rome, Italy; 2CREA, Research Centre for Forestry and Wood, Viale Santa Margherita 80, IT-52100 Arezzo, Italy; 3UNISS, Department of agriculture, NuoroForestrySchool, Via C. Colombo 1, IT-08100 Nuoro, Italy

**Keywords:** terrestrial laser scanner, lidar, forest structure, forest biodiversity, voxelization, spatial prediction

## Abstract

The understory is an essential ecological and structural component of forest ecosystems. The lack of efficient, accurate, and objective methods for evaluating and quantifying the spatial spread of understory characteristics over large areas is a challenge for forest planning and management, with specific regard to biodiversity and habitat governance. In this study, we used terrestrial and airborne laser scanning (TLS and ALS) data to characterize understory in a European beech and black pine forest in Italy. First, we linked understory structural features derived from traditional field measurements with TLS metrics, then, we related such metrics to the ones derived from ALS. Results indicate that (i) the upper understory density (5–10 m above ground) is significantly associated with two ALS metrics, specifically the mean height of points belonging to the lower third of the ALS point cloud within the voxel (HM_1/3_) and the corresponding standard deviation (SD_1/3_), while (ii) for the lower understory layer (2–5 m above ground), the most related metric is HM_1/3_ alone. As an example application, we have produced a map of forest understory for each layer, extending over the entire study region covered by ALS data, based on the developed spatial prediction models. With this study, we also demonstrated the power of hand-held mobile-TLS as a fast and high-resolution tool for measuring forest structural attributes and obtaining relevant ecological data.

## 1. Introduction

Widespread knowledge of three-dimensional forest structures is essential for effective forest planning and management, particularly in dealing with biodiversity and wildlife communities. Features associated with forest structure are important to understand and prospect the wildlife species presence and thus the habitat suitability [1,2]. The understory has significant implications not only for forest health but also for wildlife habitat, nesting, and food resources [3,4]. Besides, understory vegetation influences forest ecosystem processes, such as overstory regeneration, carbon sequestration, soil nutrient cycling, erosion, and fire intensity, and spreads in a variety of ways [3,4,5]. Given its importance, precise information on understory spatial distribution, particularly across wide areas, is becoming increasingly crucial in the forest planning and management context [1,2,6].

Both passive and active remote sensing data have been recently used to gather structural attributes of forest environments over large areas. Moreover, traditional surveys have frequently been employed as ground truth to provide direct physical mensuration of selected variables related to vegetation structure, which are frequently defined by general traits without considering the three-dimensional complexity of the forest structure [7]. Obtaining three-dimensional information on vegetation structure, on the other hand, requires the use of active sensors, such as Aerial Laser Scanning (ALS) and Terrestrial Laser Scanning (TLS) [1,8,9,10,11,12,13,14].

TLS point clouds are well suited for describing understory vegetation structure [13,15,16], as well as calibrating ALS-based models [17,18]. Despite being limited to surveying smaller areas (i.e., about one hectare as a maximum, following an intensive field effort), TLS technology allows a millimeter-level reconstruction of three-dimensional forest stand structure while avoiding the problems associated with canopy occlusion that affect ALS [15,16,19]. Mobile-TLS, on the other hand, provides fast and reliable forest stand measurements across larger areas, especially when compared to static-TLS [13,20]. However, under particular conditions, mobile-TLS alone is unable to assess the tree top height, such as in forest stands where the total tree heights exceed the laser range [12].

An integration of both ground-based and airborne-based measurements can provide high-resolution, detailed information for all forest layers. Few studies have focused on the opportunities resulting from the integration of TLS with ALS (e.g., [21,22,23]), especially when carried out in complex Mediterranean environments (e.g., [24]).

Other authors [19] have integrated static-TLS metrics with the ones derived from a full-waveform ALS sensor under a regression approach. They obtained promising results, although the experiment was carried out using a reduced number of samples per forest type (about three to four plots). Considering the complexity of Mediterranean forests, this number of samples does not suffice to quantify actual relations between ALS and TLS metrics. Besides, the use of static-TLS in forest inventory represents a limitation. The advantages of static-TLS, providing precise forest mensuration, are hindered by major drawbacks such as the extra time needed for data gathering and processing, along with highly professional skills requested. For these reasons, the use of this technology among forest technicians is quite limited. Mobile-TLS can substantially reduce such constraints.

A recent experience [14] provided a reliable solution for ALS and mobile-TLS data co-registration, developing a new GIS software (ForeSight^®^) that uses a geometric features recognition algorithm for point cloud alignment, with the ultimate goal of characterizing the different forest stand structures from a three-dimensional perspective. Their findings indicated that combining ALS and TLS approaches broadens the opportunities for obtaining highly detailed forest structural features, providing a powerful alternative to traditional techniques while advancing applications in forest management, particularly in complex Mediterranean environments.

This work aims to provide a modeling approach for gathering spatially detailed information on the three-dimensional distribution of forest understory by using mobile-TLS and ALS data. TLS and ALS metrics were processed using a voxel-based approach to quantify understory presence and density in two separate understory forest strata: Lower understory (2–5 m) and upper understory (5–10 m) [13]. Conventional forest plot data have been collected to validate the ability of TLS metrics to quantify understory presence. Finally, we exploited the relationship between TLS and ALS metrics to produce spatial estimates of understory presence across the entire study region covered by ALS data.

## 2. Materials and Methods

### 2.1. Study Site and Reference Data

The airborne lidar flight covered an area of 40 km^2^ inside the Sila National Park, Italy (Figure 1) (for more details, see [14,20]).

The analysis is based on 24, 15-m-radius, circular forest plots located within the study area: 12 in mature European beech (*Fagus sylvatica* L.) stands, 12 in mature black pine (*Pinus nigra* subsp. *laricio* Maire) stands. Diameters at breast height and a sample of the heights of the trees were measured in all plots. For further details on this dataset, please see [13,20]. Figure 2 presents statistics of plot-level characteristics by forest species. Field basic data are available at http://doi.org/10.5281/zenodo.3575529 [25].

### 2.2. Quantifying Understory Vegetation Densities using TLS Data

We used a GeoSLAM ZEB-REVO (GeoSLAM Ltd., Ruddington, UK) lightweight mobile laser scanner to collect TLS data (for further details, see [13,20]). The point clouds were normalized using the TreeLS R package [26]. The TLS dataset is available at http://doi.org/10.5281/zenodo.3633629 [27].

TLS data have been acquired, for all the forest plots, in an approximately square area with over 30 m side, including the circular plots. The investigated TLS 3D-space is limited, after normalization, to a parallelepiped with a 27 m side square base centered on the plot and a height of 15.5 m. This space was divided into regular cells (or voxels) of 0.5 × 0.5 × 0.25 m, using a voxel-grid approach [13], to compute vegetation presence-absence (Figure 3) and detailed vertical density profiles. Then, following the works of Aschcroft et al. [7] and Puletti et al. [13], Plant Vegetation Index (PDI) was computed for five height classes (or slices) representing distinct forest understory layers: forest floor (FF, 0.5–1 m), shrubs (Sh, 1–2 m), lower understory (LU, 2–5 m), upper understory (UU, 5–10 m), subcanopy (Sc, >10 m) (Figure 3).

### 2.3. Modeling Understory Vegetation Density Using ALS Metrics

To make ALS and mobile-TLS data comparable, all the points higher than 22 m in the ALS point cloud were removed, and a threshold of 0.5 m was applied to the normalized ALS point clouds [14]. ALS dataset, together with the TLS one, is available at http://doi.org/10.5281/zenodo.3633629 [27].

We used linear regressions to develop predictive models of the two understory layers (LU and UU). A total of 17 ALS metrics, listed in Table 1, were considered as possible predictors [14]. Only the metrics that displayed a significant Pearson’s coefficient of correlation higher than 0.7 have been considered for the final model definition.

A leave-one-out cross-validation procedure was applied. The coefficient of determination (R^2^) and the normalized root mean square error (nRMSE, [6])
nRMSE = RMSE/(y_max_ − y_min_)(1)
were both used to evaluate the model prediction accuracy.

To produce the wall-to-wall spatial prediction across all the area covered by the ALS, a 25 m resolution grid has been adopted to compute the metrics selected as input to the models for the LU and UU understory layers.

The GeoSLAM Hub (GeoSLAM Ltd., Ruddington, UK) proprietary software was used to convert raw TLS data to LAS files. All the analyses were performed in R [28]. We used R packages lidR [29] and TreeLS [26] to process ALS and TLS point clouds, respectively.

## 3. Results

Detailed and classified TLS-based PDI (Plant Vegetation Index) profiles are displayed in Figure 4 for all plots. Using voxel high (0.5 m) slices (detailed, grey dots) or understory height classes (classified, black dots), the densities are computed as the quota of ‘vegetated’ voxels within the stratum.

Smaller (DBH < 9 cm) and intermediate trees (DBH from 9 to 20 cm) contribute to understory vegetation density in the two considered layers. Using conventional forest plot data (tree frequencies by DBH class), the TLS-derived understory density values have been examined. The strength of the relationship, measured using the Spearman correlation coefficient, was highly significant (*p* < 0.0001) in both cases, with very high values (LU: 0.82 and UU: 0.88).

Adopting stated criteria, appropriate ALS metrics have been selected to predict TLS-based understory vegetation densities in each considered forest layer. Table 2 shows the selected ALS predictors with the corresponding R^2^, nRMSE, and significance values of the regression coefficients. At the end, the two resulting models contain only two variables among the 17 computed (Table 1). The first, HM_1/3_ — which has been used for modeling both LU and UU — explained the greatest amount of variability for understory vegetation cover. The standard deviation of the first third SD_1/3_ was selected only for LU modeling. Understandably, as ALS signal penetrate from above, the prediction accuracy was higher for the upper (UU) rather than for the lower (LU) layers (Table 2).

Figure 5 shows scatter plots of the observed TLS-based versus the ALS-based predicted variables, together with the residual analysis, for both lower (LU) and upper (UU) understory. LU residuals are relatively uniformly distributed, except for two extreme values. UU residuals are much closer to the zero line but display a possible unexplained curvature.

The spatial distribution maps for the entire study area based on the regression models for each considered understory height class are depicted in Figure 6.

## 4. Discussion

The amount of available space under the dominant forest canopy characterizes forest structure. It is an environmental resource not only for shrubs and young trees but also for the life cycle of insects and micromammals (e.g., bats [30]). Nonetheless, it has received limited attention as an inventory or a mapping subject.

In the US forest inventory, as an example, the understory issue has been raised since the 1970s [31,32] pointing out, as a fifth general requirement, that the procedure should “cause no disturbance to vegetation on permanent plots” and that each effort has been developed with specific adaptation to a given geographical area [33]. However, examining current inventory procedures [34], the understory represents still just a problem.

The analysis presented in this study demonstrates that, using mobile-TLS, it is possible to fill this information gap in an efficient way. The methods adopted here greatly increase the ability to estimate and predict understory vegetation density, also in quite complex Mediterranean forests. The surveyed areas are characterized by different degrees of forest structure heterogeneity, both horizontal and vertical. Hence the proposed method demonstrates strong ability in different forest structures.

It’s easy to imagine how dense understory vegetation reduces detection performance [34], particularly if we go over specific heights. That’s the reason why we suggest limiting the analysis to a box of 15 m in height, even in not dense forest structures (see Figure 3, the pine plot on the right). In addition, over that height limit, the forest vegetation cannot be defined as “understory”.

The voxelization approach used to analyze the point cloud is a widespread practice among researchers today [35]. In this study, such a downsampling procedure makes computation faster without reducing the quality of results. The adopted voxel-based technique has major constraints related to the so-called “ghost points”, a specific behavior of phase-shift TLS that makes the point cloud noisy around the edges of objects [36]. Such an effect is amplified when registering multiple scans, where false objects lead to biased results in canopy measurements. The proposed approach highlights how such drawbacks can be profitably overcome. Concerning the weather conditions, as always required when scanning in forests, it must be windless, not extremely cold, and not rainy or foggy during scan acquisition. The leaf-off conditions can be preferred due to higher visibility in upper understory (no-leaf disturbance).

In this study, also ALS data demonstrate a unique capacity to characterize understory structures at a high and broad spatial resolution. The selected ALS metrics allow for mapping the spatial distribution of the forest understory density measured using mobile-TLS with good overall accuracies. Few metrics were needed, both derived from the lower strata of the ALS point cloud, as HM_1/3_ (the average height of cloud points in the slice between 0 and 1/3 of the maximum height). Expectedly, higher coefficients of determination and accuracy (nRMSE) were found in the upper, rather than in the lower understory layer as, reasonably, the former can be better detected by ALS. Moreover, due to the relatively high ALS pulse density, prediction of ALS-based understory vegetation density was not affected by the canopy cover presence in the two studied forest types, even if in areas with higher overstory canopy densities and smaller canopy gaps (as for the mature beech forests) laser pulses are commonly impeded to reach the understory. This represents a significant result since the possibility of using remotely sensed information at broader spatial extents is critical to characterizing the forest structure and exploring wildlife habitat ([1,6]).

From a management perspective, the understory space investigated in this work, in particular UU, refers to the space of renewal, which is critical to ensuring forest cover over time. Despite its ecological importance, this forest structural trait is not traditionally measured in forest inventories, mainly because it is not easily measurable in the field.

The ability to obtain information on understory vegetation density over broad spatial extents remains a critical issue. Spatial distribution maps produced using the proposed methodology could be helpful to fill this gap for environmental management, particularly for preserving forest biodiversity and averting forest fires [37]. The accuracy of obtained predictions agrees with previous studies ([1,18]). In particular, Wing et al. [6], at a resolution of 40.5 m^2^, estimated understory cover using regression models, finding nRMSE similar to the ones reported in the present work. Our study further suggests that discrete ALS can be used to estimate understory vegetation density with similar errors also at lower resolution (i.e., about 700 m^2^).

## 5. Conclusions

In this work, a method for mapping understory vegetation in Mediterranean forests through the combination of mobile-TLS and ALS data has been presented.

Findings from this work indicate that the proposed approach is effective in providing a reliable and efficient alternative to traditional measurement techniques. Methods based on high-resolution optical imagery obtained from sensors mounted on satellites, airplanes, or drones can be used only when the canopy cover is not dense, while ALS has a greater canopy penetration ability. Field surveys, such as transects or quadrats [38], can be used to collect detailed data on understory density, but these procedures are time-consuming and not suited for large area measurements. Mobile-TLS speeds up the surveys [39].

The inclusion, in forest inventories, of information derived from three-dimensional data is proposed as an important step to support forest management and planning. In the last decades, the use of both ALS and TLS in forest mensuration practices has increased. However, it is still at a research level. High instrument costs and the requirement for well-trained personnel limit the widespread use of TLS technology in the forest sector. The obtained results can be useful for a comprehensive understanding of the forest ecosystem and to support effective forest planning and management decisions.

## Figures and Tables

**Figure 1 sensors-23-00511-f001:**
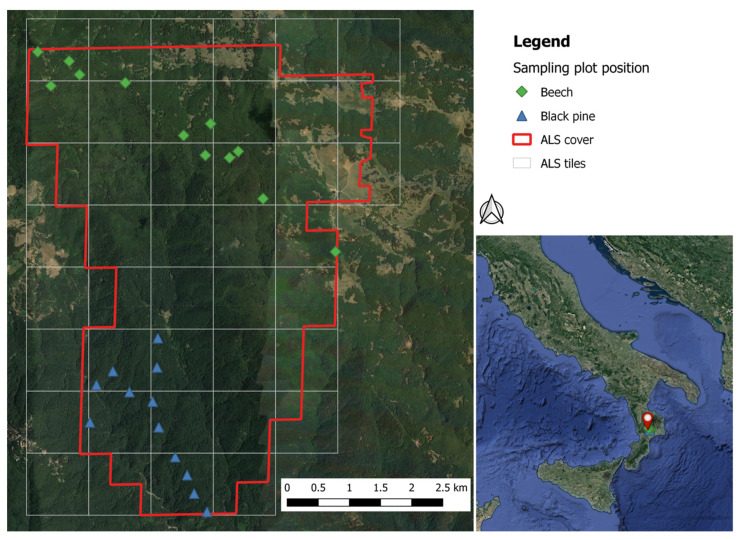
Plot locations and ALS tiles of the study area.

**Figure 2 sensors-23-00511-f002:**
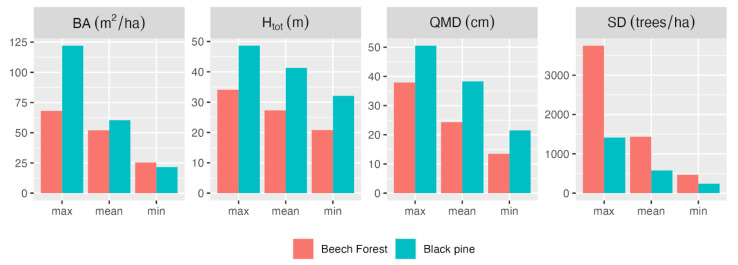
Summary plot-level attributes by dominant forest type: basal area (BA), total tree height (H_tot_), quadratic mean diameter at breast height (QMD), stem density (SD).

**Figure 3 sensors-23-00511-f003:**
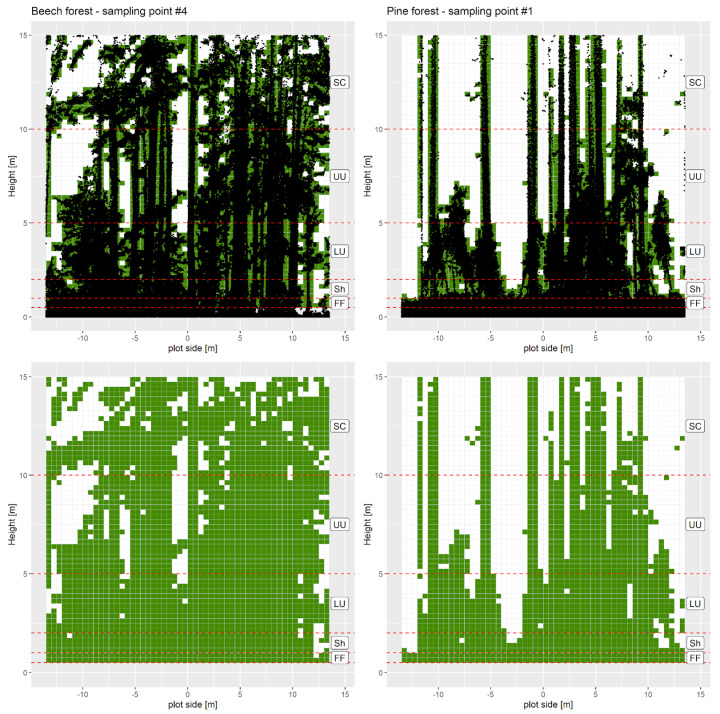
Vertical slice of the TLS–3D space surrounding two example plots. The slice is 10 voxels thick, cut along the central line. Black dots in the upper graphs are cloud points, while in the lower ones, only vegetated voxels are displayed. Voxels (0.5 × 0.5 × 0.25 m) are considered “vegetation” (green) if they contain at least five points. Point cloud is clipped at a height of 0.5 m (to avoid effects from ground vegetation) and 15.5 m (due to the working range of the mobile–TLS). Horizontal dashed lines mark layers’ limits: (FF) forest floor (0.5–1 m), (Sh) shrubs (1–2 m), (LU) lower understory (2–5 m), (UU) upper understory (5–10 m), (Sc) subcanopy (10–15.5 m).

**Figure 4 sensors-23-00511-f004:**
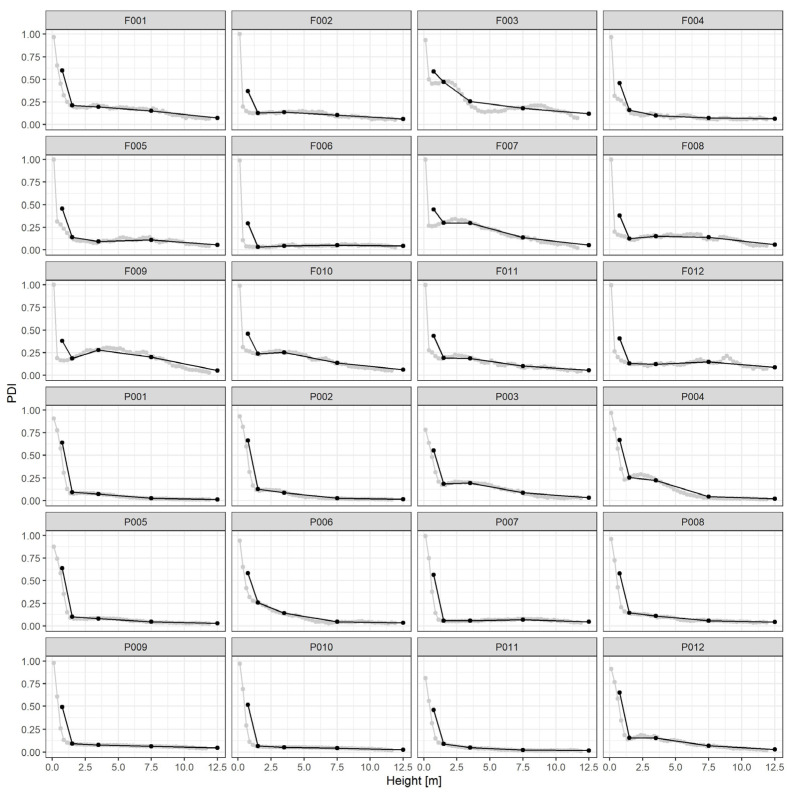
Vertical profiles of the Plant Vegetation Index (PDI), by plot, for all 24 stands. Grey dots are voxel high (0.5 m) slices averages. Black dots refer to PDI average values by height class, from left to right, they represent: FF—forest floor, Sh—shrubs, LU—lower understory, UU—upper understory, Sc—subcanopy.

**Figure 5 sensors-23-00511-f005:**
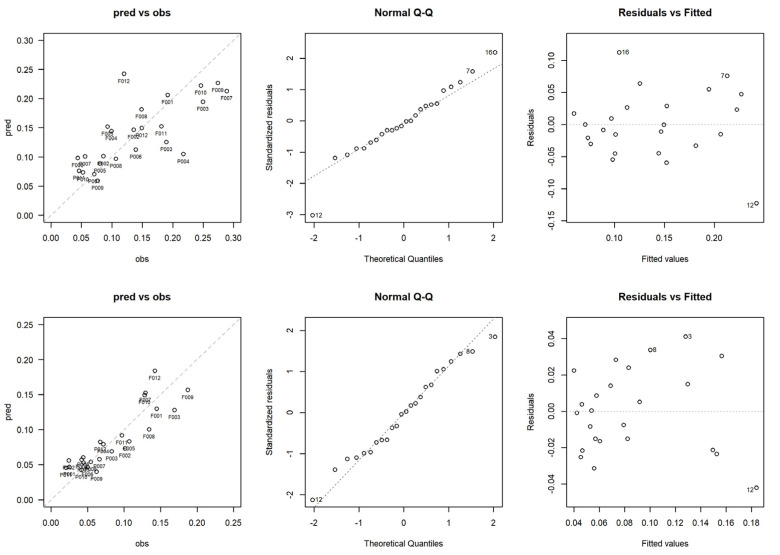
Regression graphs for the estimation of the lower (LU, first line) and upper understory (UU, second line).

**Figure 6 sensors-23-00511-f006:**
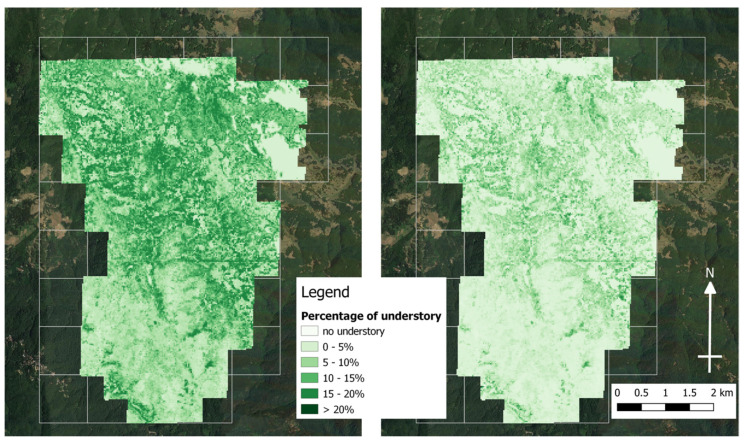
Lower (LU, on the left) and upper understory (UU, right) spatial prediction based on the regression models for the entire study area.

**Table 1 sensors-23-00511-t001:** Definition of ALS metrics. Note: To compare ALS and TLS, ALS metrics that consider the space divided into thirds (HM1, HM2, HM3, SD1, SD2, SD3, PP1, PP2, PP3) were computed up to a height of 22 m.

	ALS Metrics	Description
*Height-based* *(12 metrics)*	Mean, relative mean, and standard deviation of heights (HMEAN, RHMEAN, SDH)	The mean and relative mean heights above the ground of all first returns
	Coefficient of variation of height (HCV)	Coefficient of height variation of all first returns
	Skewness and kurtosis of height (HS, HK)	Skewness and kurtosis of the normalized heights of all first returns
	Mean and standard deviation of heights within three layers (HM_1/3_, HM_2/3_, HM_3/3_, SD_1/3_, SD_2/3_, SD_3/3_)	Mean and standard deviation of heights lower than 1/3, between 1/3 and 2/3, and higher than 2/3 of the maximum height
*Density-based* *(5 metrics)*	Percentage of points over the ground (OGP)	The number of first returns classified as no-ground over the total first returns
	Points total number (PTN)	Total number of first returns
	Percentage of points within three layers (PP_1/3_, PP_2/3_, PP_3/3_)	Percentage of points in three layers: Lower than 1/3, between 1/3 and 2/3, and higher than 2/3

**Table 2 sensors-23-00511-t002:** Leave-one-out cross-validation results obtained for the prediction of understory vegetation density in the two considered layers (LU and UU).

**Lower Understory (LU)**				
*Adjusted-R² = 0.51; nRMSE = 20%*				
Metric	Estimate	Std. Err.	*t* value	*p*-level
HM_1/3_	0.025	0.028	1.86	0.077
SD_1/3_	0.029	0.013	1.17	0.253
**Upper Understory (UU)**				
*Adjusted-R² = 0.77; nRMSE = 13%*				
Metric	Estimate	Std. Err.	*t* value	*p*-level
HM_1/3_	0.029	0.003	8.77	< 0.001

## Data Availability

Data presented in this study are available at the links mentioned in the text or on request from the corresponding author.

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
