# Peer review of "Mapping Understory Vegetation Density in Mediterranean Forests: Insights from Airborne and Terrestrial Laser Scanning Integration"

_sensors, 2023, doi:10.3390/s23010511_

Round 1
Reviewer 1 Report
The amount of available space under the dominant forest canopy cover characterizes forest structure. It is an environmental resource not only for shrubs and young trees but also for the life cycle of insects and micromammals.
For this manuscript, the introduction explains the advantages and disadvantages between TLS and ALS in detail, and through the comparison of existing work, the authors highlighting the advantages of combining TLS and ALS in the Mediterranean complex terrain. This is also the biggest innovation of this article.
In the experiments, the location of the experiment, the detail parameters of the equipment, and the data are clearly described. The experimental data prove the superiority of this algorithm.
This work is a nice application-oriented research, only a few typos need to be corrected, such as at line 118, two ‘, ,’ are typed. This manuscript can be published on the sensors.
Author Response
We are thankful for the reviewer positive assessment of our work as a valuable contribution to the field and suitable for publication in Sensors. We have also carefully
reviewed the manuscript to ensure that all errors have been corrected, following the reviewer suggestion.
Reviewer 2 Report
This paper "Mapping understory vegetation density in Mediterranean forests insights from airborne and terrestrial laser scanning integration" the authors used terrestrial and airborne laser scanning 16 (TLS and ALS) data to characterize understory in European beech and black pine forests in Italy. 17 First, we compared understory structural features derived from traditional field measurements with 18 TLS metrics, then, we related such metrics to the ones derived from ALS.
I consider this paper needs to be improved through major revisions. Check my following suggestions to further improve the paper.
1. Make extensive English revisions. Section 2, 3 have some of the longer sentences, break those sentences to increase the readability of the manuscript.
2. Section 2 lacks the analytical details.
3. Provide furthermore reasoning on the performance evaluation section 3.
4. Section 5 of conclusion needs to be extended.
5. Add some references from 2022.
Author Response
reply in the letter
